# Determinants of Bank Closures: What Ensures Sustainable Profitability in Mobile Banking?

**Soohyung Cho [1], Zoonky Lee [1], Sewoong Hwang [2] and Jonghyuk Kim [2,*]**

[1] Graduate School of Information, Yonsei University, 50, Yonsei-ro, Seodaemun-gu, Seoul 03722, Republic of Korea
[2] Division of Computer Science and Engineering, Sunmoon University, 70, Sunmoon-ro221beon-gil, Tangjeong-myeon, Asan-si 31460, Republic of Korea
[*] Correspondence: jonghyuk@sunmoon.ac.kr; Tel.: +82-41-530-2266

**Abstract:** Owing to the recent increase in mobile banking customers, studies exploring self-service channels and customer responses as distribution channels in the retail banking industry are also rapidly expanding. Moreover, with the emergence of big data and a series of artificial intelligence (AI) technologies, customer pattern analysis using deep learning, insurance fraud prevention, software development and various types of blockchain-based FinTech technologies, offline banks are disappearing. Accordingly, many commercial banks are attempting to find technological alternatives. However, maintaining a profitable bank branch is a crucial factor in the relationship between service quality and customer satisfaction because excellent service quality prevents existing customers from leaving. This study sought to quantitatively prove the change in customer service quality and profit due to the introduction of technology in the financial industry. We microscopically compared the effects between bank branch closures and changes in net profit using a time-series analysis. Specifically, we quantitatively analyzed actual customer attrition behavior with a time-series analysis across the three quarters before and after the closure of 88 branches of major commercial banks in South Korea in the Seoul metropolitan area and nearby cities. The findings prove that branch closures and multi-channel effects in the financial sector are gradually being resolved through immediate technology acceptance, contrary to popular concern.

**Keywords:** mobile banking; FinTech; bank closure; TAM; innovation resistance; self-service channel; time-series analysis; wear-in time; GIRFs; GFEVD

## 1. Introduction

A notable recent trend in financial consumption is the increase in services using multiple channels. Owing to the development of information technology (IT), the multi-channel financial distribution milieu is now providing services to customers at a very high speed and is competitively entering consumers' lives. New types of financial products and channels are constantly emerging. According to the last World Economic Forum, we are now in the fourth industrial revolution with the convergence of artificial intelligence (AI), the internet of things (IoT), and big data, and these innovations are already part of the financial industry. In addition, the financial information technology convergence sector, which uses AI and big data technology, is rapidly developing. However, it is not easy to determine the actual situation of the online and mobile banking fields and their impact on bank profit creation due to the data acquisition problems and personal information protection problems experienced by large commercial banks [1].

Over the past 30 years, technological innovations have impacted how banks operate and serve their customers. To keep pace with evolving FinTech technology, banks have also developed novel, innovative financial products and introduced competitive IT financial products, such as transaction processing and risk management systems. From a customer's

perspective, improved banking technology can provide significant convenience. However, this does not imply that all customers now have a better impression of banks in terms of their convenience. If a nearby branch is closed, customers unfamiliar with mobile devices will naturally move to another nearby bank or lose their enthusiasm for purchasing financial products. As shown in Table 1, as of November 2021, the number of unmanned branches of South Korea's five major commercial banks is approximately three times greater than the number of manned ones. This difference is approximately 1.6 times higher than that from the same period in 2016, five years before [2].

**Table 1.** Number of manned vs. unmanned branches of South Korea's five major commercial banks (November 2021).

| Commercial Bank | Manned Branch | Unmanned Branch |
|---|---|---|
| Shinhan Bank | 784 | 2589 |
| Kookmin Bank | 925 | 3060 |
| KEB-Hana Bank | 624 | 2524 |
| Woori Bank | 792 | 2163 |
| NH Bank | 1118 | 2252 |

Improving customer convenience does not simply mean predicting a rosy future where offering a variety of technology options will increase customer loyalty and competitiveness and ultimately improve a bank's performance. Due to the development of finance-IT, people continue to be marginalized, which must be addressed. Since this is also a problem directly related to not only a bank's services but also its profits, more empirical research is needed. As seen in Table 2, South Korea's five largest commercial banks have been extremely close to their branches for the past five years, except for 2018. In addition, in 2021, the difference between the increase and decrease in branches was 268—the largest drop since 2017. This may directly or indirectly result in the problem of technology acceptance and innovation by a bank's internal employees and external customers [3]. Hence, we need to analyze the factors of innovation acceptance, diffusion, and resistance.

**Table 2.** Number of establishments and closures of manned branches of South Korea's five major commercial banks (November 2021).

| Year | Establishment | Closure | Diff. |
|---|---|---|---|
| 2017 | 127 | 244 | −117 |
| 2018 | 71 | 60 | 11 |
| 2019 | 63 | 71 | −8 |
| 2020 | 25 | 88 | −63 |
| 2021 | 8 | 276 | −268 |

This paper cannot detail why the research target bank provided us with data because this information is related to the bank's trade secrets. However, in principle, environmental change needs to occur; along these lines, the development of the FinTech industry is significantly changing financial customer services, and banks are, accordingly, trying to find more fundamental technological alternatives by applying AI-based technology. In other words, the project carried out by our researchers and the target bank's mutual needs was the decisive factor in obtaining the bank's sensitive data. This also speaks to a practical connection with existing studies on the needs of customers inside and outside the bank, which we detail in our literature review, and the active response to operating profit, which is the focus of our study.

Specifically, this study aimed to examine the development of self-service channels and customer responses to empirically investigate the impact of multiple channels on bank profits. Using data from large commercial banks, this study scrutinized financial product revenue corresponding to closed branches and nearby similar/integrated branches in a

time series by customer age group. Hypothesis setting and testing were carried out using the existing research model.

This paper comprises four sections. In Section 2, we address research related to the existing financial field, multiple channels, services, and efficiency. First, we trace the development of a self-service channel and customer responses. We explore the impact of increasing banks' self-service channels, customer responses to multi-channel services, responses that appear differently depending on customer groups, technology acceptance models and customer responses, and the diffusion of innovation and resistance to innovation. Second, in terms of research on profitability and banks' service levels, we look at the methodology to empirically analyze profitability and the effect of commercial banks on customer service. In Sections 3 and 4, we set up the experimental design for our empirical analysis and subsequently outline the demographics of our data, explain the derived variables, and detail each variable and acquisition method. Next, we establish a hypothesis and statistically test it. In addition, we interpret the statistical verification and empirically explain the meaning of the data and the verification results. Section 5 summarizes the study and outlines its implications and limitations.

## 2. Background

### 2.1. The Development of a Self-Service Channel and Customer Responses

The reason for reviewing technology-based self-service channels of retail finance in the present study is that they are involved in the decline in the number of bank branches, the related increase in mobile customers, and the rise or decline of bank products. We review the importance of technology-based self-service channels as a means of distribution in the retail finance sector and customer responses. A prime example of a technology-based self-service channel is mobile banking. Mobile banking services are the most technology-intensive and new-trend services implemented by commercial banks [4]. Many studies have shown that an increase in bank revenue and customer self-service channels has a sharp effect [5,6]. In other words, self-service technology has the positive outcome of allowing customers to control the provision of services in a way that better meets their needs. In addition, self-service channels can improve control and perceived quality of service, so one can expect to benefit from service interactions. To this end, it is necessary to confirm prior research on how the psychological reactions of customers affects the sales channels of banking products, and to explore human acceptance of new technologies.

In terms of looking at service channels from another angle, let us discuss multi-channel services; these services can be subsumed under a customer's experience, including all factors in the physical offline world and the online virtual path for the quality or service of various channels. In addition, multi-channel services provide a way to increase customer value and enhance human convenience. However, these are not always positive aspects. Multiple channels can confuse customers, especially middle-aged and older adults who are not used to mechanical complexity and can feel as though they are being inconvenienced [7]. For example, they feel very uncomfortable when using new products or channels because their attention and cognitive abilities are significantly lower than those of youth when making decisions. On the other hand, in the case of young adults, when a new product is released, multiple channels serve to spread news about the product. In other words, age is the most important variable in researching new channels. According to technology diffusion theory, the acceptance of new technologies is rooted in the unique context in which they emerge; notably, this idea is also stated by the technology acceptance model (TAM). As shown in Figure 1, the TAM provides a metric that makes it easy to measure the most fundamental factors in the acceptance of numerous new technologies by customers or consumers [8]. This study focuses more on a bank's profit structure and the reduction of branches due to the development and spread of new channels, such as mobile banking and customers' responses. To this end, it is necessary to investigate how a bank's profit structure has changed and how it will evolve in the future through interactions with customers, rather than evaluating and verifying its impact in the existing industry.

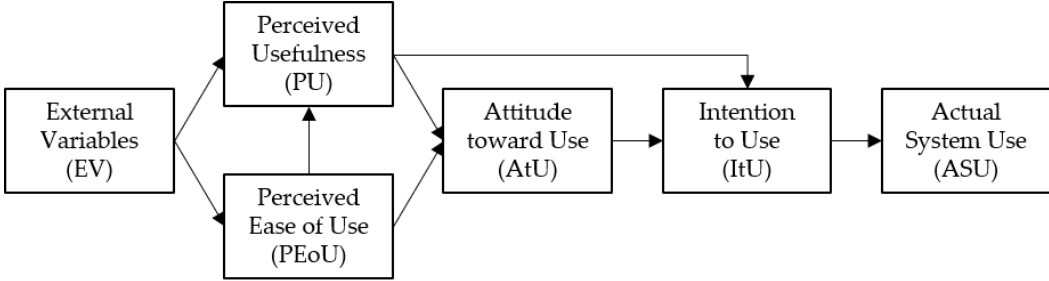

**Figure 1.** TAM (Technology Acceptance Model).

Another concept used to understand customer responses is technology readiness (TR); proposed by Parasuraman (2000), this concept highlights the differences in individual biases toward techniques used to achieve a specific goal [9]. That is, TR is composed of inconvenience and instability, along with active variables, such as positivity and innovation. TR TAM is commonly referred to as the technology readiness acceptance model (TRAM). TRAM combines the perceived usefulness or ease of the TAM with anxiety variables, positivity, and innovation to explain a person's disposition in a more sophisticated and detailed manner [10]. Strong resistance to new technologies stifles innovation. In contrast, the theory of institutionally based trust is grounded in the effect of customer reactions. According to research on channels in existing banks, innovative thinking and customers' ongoing use of new technologies can be expressed as trust or loyalty. Trust is the most favored virtue, especially in the financial sector. In other words, trust in the technology of online and mobile banking implies trust in the system; this belief is expressed in consumers' satisfaction with new technology [11]. Currently, trust theory structurally constitutes the sub-variables of situational normality and institutional assurance. Prior studies on innovation diffusion and innovation resistance are also needed. Innovation diffusion is a theory asserted by Rogers (1995) and refers to the process of not only accepting new ideas, methods and other products or services in recognizing new technologies by individuals or groups, but also all processes of informing and disseminating them. The sub-factors of the perceived features of an innovation include relative advantage, suitability, complexity, and availability [12]. Other factors, such as instability and discomfort, are connected to the idea of innovation resistance. This refers to any act of refusing to change or accept new things from the current state. As shown in Figure 2, innovation resistance is a multi-dimensional concept that highlights cognitive and emotional behavioral factors; it acts continuously throughout the process of change, not just at one specific stage in that process [13].

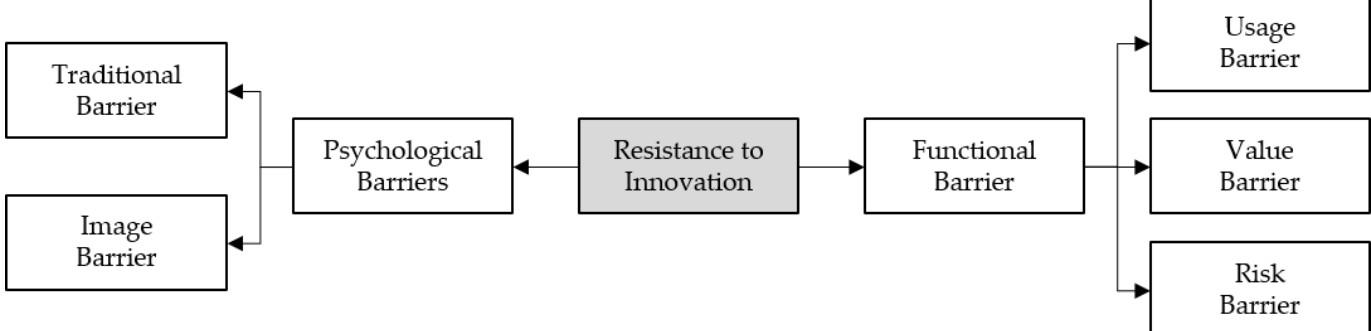

**Figure 2.** Innovation resistance.

The ideas of innovation diffusion and innovation resistance are mainly used in the financial field because of their complexity and customer characteristics. Likewise, the opportunity to recognize and use mobile banking services in advance or to observe them through communication with people who are using them can sufficiently lower resistance to online services [14].

## 2.2. FinTech Development and Smart Financing

Owing to the rapid spread of mobile devices, many consumers now engage in non-face-to-face banking, and face-to-face channels in the banking industry (which are traditional business methods) are gradually disappearing. Recently, mobile banking-based services have expanded, and customer visits to branches due to simple transfers have gradually decreased. In addition, smartphones continue to expand consumers' online access while replacing various other banking services [15]. Moreover, the development of the mobile banking market, along with consumers' need for convenience, enables consumers—especially youth—to enjoy near-perfect financial services on mobile devices through their smartphones [16].

Advanced technology is grafted onto finance, which creates a fusion synergy effect, leading to FinTech development. The evolution of this industry has occurred as part of customer service, but due to new technological developments, bank branches have become structures that interfere with profits. Thus, the traditional financial sector is attempting to morph into a new financial industry [17]. In recent years, a series of AI technologies—such as big data, customer pattern analysis using deep learning, the development of insurance fraud prevention software (S/W), and various kinds of FinTech technologies rooted in block chains—has emerged and banks are disappearing. Even machines such as automatic teller machines (ATMs) at banks are becoming less necessary, and the way bank branches operate is changing significantly. For example, there is a fundamental question about whether customers can ask questions and receive answers through a real-time chatbot system in the absence of a live agent [18,19]. Many commercial banks are trying to find technological alternatives to answer these questions.

## 2.3. Bank Profitability Analysis and Financial Services

Profitability indicators for commercial banks cannot be easily identified unless information is provided by internal personnel. This means that there are very few previous studies dealing with improvement indicators for bank profits [20]. We decided that the most important factor in analyzing the profits of closed and nearby branches is to calculate individual sales and contribution profits to the bank, rather than the total sales of stores. Thus, it was necessary to trace the relationship between profitability improvement and banking service quality in the South Korean banking industry [21]. According to a study using data from large commercial banks, regression analysis was used to verify whether fluctuations in market interest rates have a significant impact on bank profitability, and to develop a method for estimating the average asset and debt maturity of banks [22]. According to another study, profitability inside and outside a bank was also derived from the results of return on asset (ROA), return on capital (ROC), and value-added return on total assets [23]. To examine the causes of bank closures, the determinants of a bank's business performance were identified through data from 22 commercial banks collected over approximately four years. Some studies have found that the profitability of commercial banks is influenced in a complex way by a bank's asset portfolio and productivity variables. However, the units of analysis for this kind of study were mostly store units. In other words, it is not easy to find a unit of analysis for each customer [24].

A bank's service quality is directly tied to its profits and survival and is a vital factor in securing a bank's differentiated competitive advantage. Service quality can reduce the cost of acquiring new customers by retaining existing ones, making them feel satisfied with the services they receive, increasing their purchases, and making the bank recognize the value of the services provided [25]. In general, the higher the perceived service quality, the higher the customer satisfaction with the transaction relationship. Many studies have empirically investigated the causal link between service quality and customer satisfaction for bank customers, and have shown that it is reasonable to view service quality as a leading variable in customer satisfaction. As for the connection between service quality and customer satisfaction, many studies have established that excellent service quality can prevent the departure of existing customers and attract new ones [26]. In addition,

high service quality—rooted in interactions between customers and bank employees—can positively influence customer satisfaction. In other words, the maintenance of bank branches is a critical factor in a bank's profits. A study on bank customers found that cross-purchase intention rose as customer satisfaction and perceptions of convenience increased [27].

## 3. Methodology

The three basic classification criteria of this study were as follows: First, we defined customers who use online or mobile banking services as "active customers" and we classified those who do not as "non-active customers." In other words, we defined the standard of an active customer as one whose net profit in the number of online or mobile banking transactions exceeds zero from the perspective of customer data. We divided the second breakpoint into points before and after a bank's branch closure date. We used August 2020 to February 2021 as the reference point for bank closures when many bank branch closures occurred. We also employed data from April to December 2019 for "time before closure" and April to December 2021 for a "post-closure" stage. We also compared the net profit per customer (pre-tax profit) and the number of product transactions for closed bank branches and similar bank branches adjacent to closed ones in both time periods. Similar nearby bank branches are those nearest to closed bank branches, which are very similar to closed ones in terms of the scale of operating assets; we are referring to bank branches where closed ones are ultimately absorbed and integrated. The final classification criterion in this study was generational classification. In other words, based on the general retirement age of South Korea, we compared the banking services of diverse channels by dividing people into two age groups. More specifically, we created two age groups based on the international standard age of 55 used in many studies [28–30]. Accordingly, the working-age population was defined as youth and young people, which comprised individuals under the age of 55, and the retired population was defined as middle-aged and elderly people, who comprised individuals over the age 55.

We conducted our analysis based on real bank customer data from 44 closed bank branches and 44 nearby similar ones during the three quarters before and after the bank branch closures. The variables were customer age, branch classification, activation status, number of online or mobile users, number of other product transactions, annual net profit, and the number of types of banking products. In other words, we drew objective conclusions by carrying out a traditional, statistically based empirical analysis that establishes hypotheses through existing research and tests them through statistical analysis. The purpose of this study is to empirically identify the impact of the recent development of AI technology and the resulting multi-channel increase in the banking business on customer service and bank profits through vast, valuable bank data. To this end, we performed data pre-processing for the hypotheses, along with demographic information on actual commercial bank data in South Korea. In addition, we conducted quantitative tests using the Oracle Virtual-Box, a virtual machine, and SAS Cloud, an open-source statistical package.

### 3.1. Data Collection and Descriptive Analysis

We obtained data before and after the closure of a commercial bank in South Korea based on the point in time when branch closure occurred on a large scale. Significant branch closures happened between August 2020 and February 2021. Based on the time of closure, there are two reference periods: the 3rd quarter preceding the closure (1 April~31 December 2019) and the 3rd quarter after closure (1 April~31 December 2021). We set the target variable based on the annual revenue generated by each customer over 18 months. We divided the total revenue into revenue generated from online banking, mobile banking, and bank counters, which are offline transactions. We masked customer information with an identifiable ID to ensure the protection of personal information. However, since age was the most important piece of information in this study, we divided the sample into the two groups listed above, namely youth and young people (individuals under age 55), and

middle-aged and elderly people (individuals over age 55). In this study, we included only individual customers, excluding corporations.

We targeted 44 branches at closed sites of the Seoul metropolitan area and the Gyeonggi-do region. We also defined nearby and post-integrated branches as 44 nearby branches with operating assets similar to those of closed branches. The average number of customers per branch is approximately 5000; after integration, the customer information of the closed branch is transferred to the nearby one, which is the integrated branch. However, for the purpose of this study, we assumed that the customers information for a closed branch (even after 2021) would be the existing closed branch, not the nearby and post-integrated branch to which their customer information had already been transferred. In addition, we differentiated the case of 0 from the case of non-zero based on the number of annual transactions to indicate whether online and mobile banking customers were active. We also quantified the net profit generated by individual units by branch, not only by dividing it into the net profit of online, mobile, and bank teller transactions, but also by dividing it into "before" and "after" branch closures. Table 3 outlines the data pre-processing status.

**Table 3.** Data classification and pre-processing.

| Variable | Requirement |
|---|---|
| Customers' ID (Pre-masked) | Primary key (unique value) has only one value |
| Age | 0 = under 20 s, 1 = 30 s, 2 = 40 s, 3 = 50 s, 4 = over 60 s |
| Age group | 0 = including and under 55, 1 = over 55 |
| Customer classification | 0 = PB, 1 = PRB, 2 = PfB, 3 = PsB, 4 = SME |
| Branch classification | 0 = closed branch, 1 = nearby branch from closed one |
| Branch closure date | Nearby branch applies null value |
| Number of online transactions per year | Distinct var. before/after closure: create additional var. |
| Number of mobile transactions per year | Same as above |
| Number of online & mobile transactions | Same as above |
| Online banking activation | If the number of annual banking transactions is greater than 1, otherwise 0 |
| Mobile banking activation | Same as above |
| Gross annual net income | Distinct var. before/after closure: create additional var. |
| Online banking net income | Same as above |
| Mobile banking net income | Same as above |
| Other accrued net income | Same as above |
| Number of financial products annually | Same as above |

As of 2022, based on the total number of customers, 227,992 for closed branches and 240,112 for nearby ones, the total number of customers analyzed was 468,104. Demographic indicators are presented in Table 4. According to the simple number of customers, about 70% of all customers are youth and young people, while 30% are middle-aged and elderly. Looking only at the number of online and mobile transactions out of all transactions, in the case of youth and young people, the number of post-closure transactions rose by around 66.2% from 225 to 374 per person. On the other hand, among middle-aged and elderly people, the number of post-closure transactions increased by approximately 181.1%, from 37 to 104 per person. This implies that although the number of simple transactions among middle-aged and elderly people was smaller than that of youth and young people, the rate of spread increased much faster.

There was a significant increase in online and mobile banking activation rates during the post-closure period. In particular, for middle-aged and elderly people, the activation rate—which was only 10.2% before closure—increased to 54.8%. Nevertheless, in terms of net profit, it was 31.6%, not keeping up with face-to-face profits (68.4%). In addition, as for the total annual net profit by individual, although middle-aged and elderly people are larger in terms of absolute amount, the percentage of net profit generated by using online and mobile devices is more than twice as high among youth and young people than

the face-to-face net profit. Conversely, for middle-aged and elderly people, the net profit derived from face-to-face occurrences was more than twice as large.

**Table 4.** Descriptive statistics.

| Division | | Period before Branch Closure | | Period after Bank Closure | | Change (%) |
|---|---|---|---|---|---|---|
| Number of customers | Youth and young people (Including and under the age of 55) | 311,710 | 69.2% | 322,048 | 68.8% | |
| | Middle-aged and the elderly people (Over the age of 55) | 136,768 | 30.8% | 146,056 | 31.2% | |
| | Total | 448,478 | 100.0% | 468,104 | 100.0% | |
| Activated customers, activation rate of online & mobile banking (persons, %) | Youth and young people | 201,676 | 64.7% | 294,996 | 91.6% | 26.9% |
| | Middle-aged and elderly people | 13,950 | 10.2% | 80,039 | 54.8% | 44.6% |
| | Total | 215,626 | 48.1% | 375,035 | 80.1% | 32.0% |
| Annual online & mobile banking transactions (cases) | Youth and young people (Per person) | 45,193,060 225 | | 110,328,492 374 | | 66.2% |
| | Middle-aged and elderly people (Per person) | 518,186 37 | | 8,324,024 104 | | 181.1% |
| | Total (Per person) | 45,711,246 262 | | 118,652,516 478 | | 82.4% |
| Gross annual net income (per person, KR Won) | Youth and young people | 455,708 | | 477,092 | | |
| | Middle-aged and elderly people | 468,256 | | 503,572 | | |
| Net income generated from online & mobile banking per year (per person, KR Won) | Youth and young people | 306,692 | 67.3% | 339,690 | 71.2% | |
| | Middle-aged and elderly people | 63,682 | 13.6% | 159,128 | 31.6% | |
| Annual face-to-face (counter) net profit (per person, KR Won) | Youth and young people | 149,016 | 32.7% | 137,402 | 28.8% | |
| | Middle-aged and elderly people | 404,574 | 86.4% | 344,439 | 68.4% | |

The number of South Korean Bank A branches in Seoul decreased from 373 in 2015 to 351 in 2016; 22 branches closed. In 2020, this number was further reduced to 346, and by the end of 2021, only 321 branches were in operation. Bank A's total assets (including deposits and loans) amounted to KRW 348.524 trillion at the end of 2018, while its 2018 net profit was KRW 2.229 trillion, which rose to KRW 2.329 trillion in 2019. In 2020, due to the impact of COVID-19, it fell slightly to KRW 2.078 trillion but rose again to KRW 2.495 trillion by 2021. The average total assets and net profit per branch were KRW 450 billion and KRW 2.5 billion, respectively. Table 5 provides information on the 44 closed branches and 44 nearby and post-integrated ones in the Seoul metropolitan area and the Gyeonggi-do region for 2020 and 2021.

We conducted further empirical and quantitative verification using a time-series model to compare the increase and decrease in net profit before and after branch closure. First, we assumed that the difference in operation method between closed and nearby and post-integrated branches was a premise that could be defined as an independent variable. We defined five variables to quantify the different operating methods of the various branches: the number of customers per branch (CST), the number of secessions at closed branches (SCD), the number of operating staff per branch (STF), the branch operation and maintenance costs (including labor costs; OMC) and the real property cost (RPC), which is related to branch leasing. As a dependent variable that has a causal link with the above variables, we defined it as the individual net profit change (RVN), which is the customers' contribution to the net profit of the branch. Finally, we used branch-specific financial information as a control variable for model creation. In general, accounting indicators that can measure institutional value (such as institutional size, sales, and debt levels) are used as control variables. We employed accounting indicators as the exogenous variables, such as total assets (TAS) as an index of branch size, liquidity (LQD) as an index of current cash flow, financial leverage (FLV) as an index of the debt ratio, and the representative Korean

stock index (KOSPI). The operational definitions of each variable utilized in this study are listed in Table 6.

**Table 5.** The information about the bank branches.

| No. | Closed Branch | Nearby Branch | No. | Closed Branch | Nearby Branch |
|---|---|---|---|---|---|
| 1 | Bundang Sunae-dong | Bundang Central | 23 | Ujangsan Station | Hwagok Station |
| 2 | Teheranro, Finance Center | Seolleung Corp. Finance | 24 | Moran Station | Seongnam Central |
| 3 | Gangnam-daero Center | Gangnam Finance | 25 | Bundang Jeongja-dong | Bundang |
| 4 | Jongno 3-ga | Jongno Finance | 26 | Park Dal-dong | Anyang Finance |
| 5 | Gachon University | Seongnam | 27 | Dongtan Cheonggye | Dongtan Station Finance |
| 6 | Ingye-dong | Suwon Central | 28 | Yeokgok | Bucheon Station |
| 7 | Banghwa-dong | Banghwa Station | 29 | Seoknam-dong | Gajwa-dong |
| 8 | Bundang Top Village | Yatap Station | 30 | Gocheok Intersection | Gaebong-dong |
| 9 | Shingeumho Station | Haengdang-dong | 31 | Heukseok-dong | Sangdo Station |
| 10 | Pyeongchang-dong | Hyoja-dong | 32 | Sindorim-dong | Guro Station |
| 11 | Wonhyo-ro | Yongsan Electronics | 33 | Siheung-daero | Siheung-dong |
| 12 | Samseongyo | Daehak-ro | 34 | Changdong Station | Nowon Station |
| 13 | Dangsan Central | Yeongdeungpo | 35 | Dongducheon | Yangju Finance |
| 14 | Walkerhill | Gwangjang-dong | 36 | Galhyeon-dong | Yeonsinnae |
| 15 | Pyeongchon (South) | Pyeongchon | 37 | Malli-dong | Mapo |
| 16 | Konjiam Finance | Gwangju Finance | 38 | Ahyeon-dong | Chungjeong-ro |
| 17 | Nonhyeon Station | Nonhyeon-dong | 39 | Yangjae Hibrand | Yangjae Station Finance |
| 18 | Bangi-dong | Jamsil Finance | 40 | Sinseol-dong | Changsin-dong Finance |
| 19 | Seongsan-dong | Sangam Finance | 41 | Gangseo | Gayang Station |
| 20 | Apgujeong (West) | Apgujeong Station | 42 | Daerim-dong | Daelim Central |
| 21 | Jukjeon | Jukjeon Central | 43 | Jeongneung | Gileum-dong |
| 22 | Cheongdam Station | Cheongdam-dong | 44 | Mangu-dong | Sangbong Station |

**Table 6.** Operational definitions.

| Variables | Operational Definition | Abbreviation |
|---|---|---|
| Independent Var. | - Number of customers per branch (CST)<br>- Number of secedes in closed branches (SCD)<br>- Number of operational staffs per branch (STF)<br>- Operation and maintenance cost (OMC)<br>- Real property cost for branch leasing (RPC) | CST<br>SCD<br>STF<br>OMC<br>RPC |
| Dependent Var. | - Rate of return | RVN |
| Control<br>(or Exogenous) Var. | - Total assets for each branch<br>- Current assets each branch<br>- Debt ratio for each branch<br>- KOSPI index daily (applied simultaneously) | TAS<br>LQD<br>FLV<br>KSP |

As mentioned above, we conducted pre-processing with formed operational definitions of the variables for empirical analysis based on actual customer financial data from Bank A during the three quarters before and after branch closures. To pre-process the financial data, we performed dummy variable processing, missing value management, outlier removal, standardization for variable comparison, logarithm processing, correlation analysis between the independent variables, and variable selection through multicollinearity. Finally, we finalized the dataset for model generation and conducted a quantitative empirical analysis.

### 3.2. Hypotheses

We examined how revenue changes as a bank's customer service channels diversify. To this end, we statistically tested the difference in bank revenue (contribution per customer) by classifying not only online and mobile banking activation, but also consumer age groups. We posited that this distinction would be a useful indicator of customer dissatisfaction with bank services owing to branch closures. In other words, if there is an actual decrease in

the net profit per person for a closed branch or a significant difference compared to the increase in the net profit of a nearby branch, we can assume that this phenomenon is the effect of bank closures. We not only classified customer groups by age; we also classified customers' online and mobile banking usage activation statuses and formed hypotheses regarding the effect on the increase or decrease in a bank's net profit, as follows:

**H1a:** *For active customers who are youth and young people, regarding the changes in a bank's net profit before and after branch closures, the integration of closed branches and nearby and post-integrated ones is the same.*

**H1b:** *For active customers who are middle-aged and elderly, regarding the changes in a bank's net profit before and after branch closures, the integration of closed branches and nearby and post-integrated ones is the same.*

**H1c:** *For inactive customers who are youth and young people, regarding the changes in a bank's net profit before and after branch closures, the integration of closed branches and nearby and post-integrated ones is the same.*

**H1d:** *For inactive customers who are middle-aged and elderly, regarding the changes in a bank's net profit before and after branch closures, the integration of closed branches and nearby and post-integrated ones is the same.*

To test these hypotheses, we analyzed the customer contribution benefits for each cluster. In other words, we set the change in revenue for individual customers as the dependent variable and the activation of online and mobile banking as the group variable. We then conducted a *t*-test on two samples by age group. In addition, we combined the four hypotheses through a 2 by 2 hybrid model, as depicted in Table 7.

**Table 7.** 2 by 2 Hybrid model for H1 testing.

| | Activated Customers (AC) | Non-Activated Customers (NC) |
|---|---|---|
| Youth and Young People (YY) | **[YY-AC]** | **[YY-NC]** |
| | Closed branches vs. Nearby & post-integrated branches (Contribution to customer revenue through comparison of changes in bank net profit before and after branch closure-integration) | |
| Middle-aged and The elderly people (ME) | **[ME-AC]** | **[ME-NC]** |

We assumed that some customers would leave for other banks due to deteriorating bank customer service, such as the closure of a nearby branch, which would reduce the net profit. As seen in Table 7, the vertical plane of the hybrid model is divided into youth and young people, and middle-aged and elderly people, while the horizontal plane is divided into active and inactive customers. We compared the change in the net profit (or customers' contribution to the bank) of closed branches and nearby and post-integrated branches during the three quarters before and after branch closures based on the four quadrants, as represented by YY-AC, YY-NC, ME-AC and ME-NC.

We conducted a more empirical and quantitative verification by implementing a time-series model to reinforce the first hypothesis. In other words, we measured the customers' contributions to a branch's current net profit according to the difference in management methods between closed branches and nearby and post-integrated ones as individual net profit rose or fell. We assumed that branch closures would have a positive effect on bank profitability and therefore proposed the following hypothesis:

**H2:** *The operation method of both closed branches and nearby and post-integrated ones has the same effect on changes in a bank's net profit before and after branch closures and integration.*

We developed a 5-step, conventional time-series model to empirically test the effects of bank revenue on branch closures, specifically a vector auto-regression (VAR) primitive

time-series model along with a unit root test to secure stationarity. We also performed a Granger causality verification using covariate analysis, used generalized impulse response functions (GIRFs) to estimate the unit standard shock effect, and used a generalized forecast error variance decomposition (GFEVD) to compare final returns. The five processes are as follows. The first step involves implementing the time-series model, which creates a VAR model. The second step, the unit root test, is a method meant to secure the stationary test of the model. This eliminates unexpected fluctuations of endogenous variables (i.e., trends or seasonality patterns outside the average range), and we conducted the augmented Dickey–Fuller (ADF) test. The third step entailed implementing the Granger causality test to determine whether each independent variable had a significant response through a covariate analysis of the response variable. The fourth step involved estimating the time interval of the response to the impact (i.e., the wear-in time) by measuring the time-series response. To do this, we implemented GIRFs. Based on the estimation coefficient of VAR, we predicted that all variables in the model would react over time when applying a certain impact size to any variable in the model. In addition, we quantitatively estimated the effect of a 1-standard-deviation shock on one of the error terms for the present and future values of endogenous variables to show the speed of causality toward the final dependent variable: the rate of return. Finally, we performed GFEVD, which measures, in percentage terms, how much the lag of one variable responds to unexpected changes in another variable. To this end, we tested the sensitivities of the independent and dependent variables using the Kruskal–Wallis test. At this point, we knew which independent variable would be more sensitive to the outcome variable and produce a positive effect.

## 4. Results

### 4.1. Test for H1

H1a means that in the case of active customers who are youth and young people, the rate of change before and after the bank contribution of closed branches is not different from the rate of change before and after the bank contribution of nearby and post-integrated branches. The total population comprises youth and young people and 294,996 active customers, among which 143,692 customers belong to closed branches and 151,304 customers belong to nearby and post-integrated branches. The nature of the dependent variable is the before-and-after change in the individual bank revenues of closed, nearby and post-integrated branches. Thus, we did not use a separate a common logarithm, but set the raw data of the amount of revenue to each ratio and tested the significance of the difference in the ratio for each branch.

In the case of H1a, as shown in Table 8, the assumption of equal variance was satisfied by the F-test (F = 1.05). Hence, we used the pooled *t*-test, and the test outcome adopted the hypothesis (t = 0.2584). That is, in the case of active customers who are young people, the difference in net profit increase or decrease by branch is not statistically clear. This means that active customers who are youth and young people did not see any change in revenue due to the difference between closed branches and nearby and post-integrated ones.

**Table 8.** Two-sample *t*-test for H1a testing.

| Div. | Variance | DF | t-Value | *p*-Value |
|---|---|---|---|---|
| Pooled | Equal | 294,994 | 0.2584 | 0.248 |
| Satterthwaite | Unequal | 292,541,182 | 0.2581 | 0.248 |
| **Equality of Variance** | **Num DF** | **Den DF** | **F-Value** | ***p*-Value** |
| Fooled F | 151,304 | 143,692 | 1.05 | 0.7852 |

H1b is the case of active customers who are middle-aged and elderly, and it is assumed that the change rate before and after the bank contribution of closed branches is not different from that of nearby and post-integrated branches. The total population is middle-aged and

elderly people, and active customers, a total of 80,039, of which 39,632 customers belong to closed branches and 40,407 customers belong to nearby and post-integrated branches.

For H1b, as seen in Table 9, the F-test indicated that the assumption of equal variance was satisfied (F = 1.02). As such, we used the pooled *t*-test, and the test outcome adopted the hypothesis (t = 0.2152). In other words, in the case of active middle-aged and elderly customers, the difference in net profit increase or decrease by branch is not statistically clear. This means that active middle-aged and elderly customers did not see any change in revenue owing to the difference between closed branches, and nearby and post-integrated branches.

**Table 9.** Two-sample *t*-test for H1b testing.

| Div. | Variance | DF | t-Value | p-Value |
|---|---|---|---|---|
| Pooled | Equal | 80,037 | 0.2152 | 0.351 |
| Satterthwaite | Unequal | 80,035,245 | 0.2150 | 0.351 |
| **Equality of Variance** | **Num DF** | **Den DF** | **F-Value** | **p-Value** |
| Fooled F | 40,407 | 39,632 | 1.02 | 0.842 |

H1c is the case of youth and young people, and inactive customers; the rate of change before and after the bank contribution of the closed branch is not different from the rate of change before and after the bank contribution of the nearby and post-integrated branch. The total population comprises youth and young people, and inactive customers, so the total number is 27,052. Among them, 12,860 customers belonged to closed branches, and 14,192 customers belonged to nearby and post-integrated branches.

For H1c, as presented in Table 10, the F-test showed that the assumption of equal variance was satisfied (F = 1.10). Therefore, we used the pooled *t*-test, and the test outcome adopted the hypothesis (t = 0.3214). In other words, in the case of youth and young people, and inactive customers, the difference in net profit increase or decrease by branch is not statistically clear. This means that from the perspective of youth, young people, and inactive customers, there was no change in revenue due to the difference between closed branches and nearby and post-integrated branches.

**Table 10.** Two-sample *t*-test for H1c testing.

| Div. | Variance | DF | t-Value | p-Value |
|---|---|---|---|---|
| Pooled | Equal | 27,050 | 0.3214 | 0.482 |
| Satterthwaite | Unequal | 27,048,254 | 0.3211 | 0.482 |
| **Equality of Variance** | **Num DF** | **Den DF** | **F-Value** | **p-Value** |
| Fooled F | 14,192 | 12,860 | 1010 | 0.754 |

H1d indicates that for middle-aged and elderly people who are inactive customers, the rate of change before and after the bank contribution of closed branches is not different from the rate of change before and after the bank contribution of nearby and post-integrated branches. The total population consists of middle-aged, elderly, and inactive customers, so the total number is 66,017. Among them, 32,074 customers belong to closed branches and 33,943 customers belong to nearby and post-integrated branches.

In the case of H1d, as depicted in Table 11, the assumption of equal variance was satisfied by the F test (F = 1.06). Therefore, we referred to the pooled *t*-test, and unlike the other hypotheses, H1d was rejected (t = 15.0413). In other words, in the case of active middle-aged and elderly customers, the difference in net profit increase or decrease by branch is statistically clear. This means that for middle-aged and elderly people, and inactive customers, there is a clear difference in the change in revenue owing to the difference between closed branches and nearby and post-integrated branches. In addition,

these results suggest that customers' willingness to purchase financial products has fallen significantly because of branch closures.

**Table 11.** Two-sample *t*-test for H1d testing.

| Div. | Variance | DF | t-Value | *p*-Value |
|---|---|---|---|---|
| Pooled | Equal | 66,017 | 15.0413 | <0.0001 |
| Satterthwaite | Unequal | 66,012,851 | 13.8547 | <0.0001 |
| **Equality of Variance** | **Num DF** | **Den DF** | **F-Value** | *p*-**Value** |
| Fooled F | 33,943 | 32,074 | 1.06 | 0.822 |

The results of the first hypothesis can be expressed as the 2 by 2 hybrid model presented above. As outlined in Table 12, the YY-AC, YY-NC, and ME-AC cases are all adopted, which means that there is no difference in the rate of change before and after the bank contribution, depending on the difference between the closed branch and the nearby and post-integrated branch. The hypothesis was rejected only in the case of ME-NC; that is, middle-aged and elderly people, and inactive customers. As such, we found that the before-and-after growth rates of the bank contributions of nearby and post-integrated branches were statistically different.

**Table 12.** Result of H1 testing through 2 by 2 hybrid model.

| | Activated Customers (AC) | Non-Activated Customers (NC) |
|---|---|---|
| Youth and Young People (YY) | **[YY-AC]** CNP of Closed Brs. (=) CNP of Nearby Brs. | **[YY-NC]** CNP of Closed Brs. (=) CNP of Nearby Brs. |
| | Comparison to CNP * (* CNP: Changes in bank net profit before and after branch closure) | |
| Middle-aged and The elderly people (ME) | **[ME-AC]** CNP of Closed Brs. (=) CNP of Nearby Brs. | **[ME-NC]** CNP of Closed Brs. (<) CNP of Nearby Brs. |

We found the following through empirical analysis: the development of information and communications technology (ICT) enabled multi-channel banks, which soon led to branch closures. We found that the deterioration in service due to branch closures and the corresponding increase in bank contribution profits had little effect on youth and young people, and active customers; however, it clearly had a negative impact on inactive middle-aged and elderly customers. Our demographic analysis already confirmed that the contribution gains and losses of active customers, who are youth and young people, are larger than the impact of inactive middle-aged and elderly customers on an absolute scale. Therefore, branch closures and multi-channel effects in the financial industry are gradually being resolved by immediate technology acceptance, contrary to social concerns.

*4.2. Test for H2*

The second hypothesis compares the difference in operating methods between closed branches and nearby and post-integrated branches, and quantitatively tests the effect of an individual's contribution to the bank's return rate through a time-series analysis model. To this end, we created a VAR model. The endogenous variable for deriving the VAR model is the dependent variable RVN, and the six independent variables (already mentioned) are as follows: CST, SCD, STF, OMC, and RPC. The exogenous variables are TAS, LQD, FLV, and

the KOSPI of the branch; they act as control variables. We set up the VAR model based on several operationally defined variables.

$$
\begin{bmatrix} RVN_t \\ CST_t \\ CSD_t \\ STF_t \\ OMC_t \\ RPC_t \end{bmatrix} = \begin{bmatrix} \alpha_1 + \delta_1 t \\ \alpha_2 + \delta_2 t \\ \alpha_3 + \delta_3 t \\ \alpha_4 + \delta_4 t \\ \alpha_5 + \delta_5 t \\ \alpha_6 + \delta_6 t \end{bmatrix} + \sum_{k=1}^{K} \begin{bmatrix} \rho_{1.1}^k \cdots \rho_{1.6}^k \\ \rho_{2.1}^k \cdots \rho_{2.6}^k \\ \rho_{3.1}^k \cdots \rho_{3.6}^k \\ \rho_{4.1}^k \cdots \rho_{4.6}^k \\ \rho_{5.1}^k \cdots \rho_{5.6}^k \\ \rho_{6.1}^k \cdots \rho_{6.6}^k \end{bmatrix} \bullet \begin{bmatrix} RVN_{t-k} \\ CST_{t-k} \\ CSD_{t-k} \\ STF_{t-k} \\ OMC_{t-k} \\ RPC_{t-k} \end{bmatrix} + \begin{bmatrix} \tau_{1.1} \cdots \tau_{1.4} \\ \tau_{2.1} \cdots \tau_{2.4} \\ \tau_{3.1} \cdots \tau_{3.4} \\ \tau_{4.1} \cdots \tau_{4.4} \end{bmatrix} \bullet \begin{bmatrix} TAS_t \\ LQD_t \\ FLV_t \\ KSP_t \end{bmatrix} + \begin{bmatrix} \varepsilon_{1t} \\ \varepsilon_{2t} \\ \varepsilon_{3t} \\ \varepsilon_{4t} \\ \varepsilon_{5t} \\ \varepsilon_{6t} \end{bmatrix}
$$

In the above equation, *t* is the time difference, $\alpha_i$ (*i* = 1, 2, ..., 6) is a constant, and $\delta_i$, $\rho_{i.j}^k$, $\tau_{i.j}$ (*i, j* = 1, 2, ..., 6) are all coefficients. In addition, *k* is the length of the differentiation, i.e., the lag length, and $\varepsilon_i$ (*i* = 1, 2, ..., 6) represents the residual as white noise.

Along with the creation of the VAR model, we conducted a time-series analysis through a 4-step process and tested the significance of H2. The second process entailed the unit root test, and we carried out the ADF test to continuously perform k-order differences until a variable that did not reject H0 was created among all 44 points. At this time, we ensured that all variables showed values less than the critical value at the 5% confidence level. As Table 13 indicates, we secured stationary branches that could not be rejected in some intervals of the debt ratio variable through the second and third differences.

**Table 13.** Stationary test of endogenous variables.

| No. | RVN | CST | SCD | STF | OMC | RPC | TAS | LQD | FLV | KSP |
|-----|-----|-----|-----|-----|-----|-----|-----|-----|-----|-----|
| 1 | −22.4 | −23.6 | −18.6 | −19.3 | −14.3 | −12.3 | −20.3 | −6.5 | −6.2 | −8.5 |
| 2 | −24.6 | −25.4 | −17.6 | −18.2 | −12.1 | −13.9 | −18.7 | −9.2 | −4.7 | −7.6 |
| 3 | −23.5 | −25.2 | −19.7 | −20.8 | −9.5 | −14.2 | −19.2 | −7.5 | −3.4 | −4.9 |
| 4 | −22.4 | −22.6 | −17.6 | −18.3 | −14.3 | −11.3 | −19.3 | −6.5 | −5.2 | −7.5 |
| 5 | −24.6 | −26.4 | −18.6 | −18.2 | −12.1 | −14.9 | −19.7 | −10.2 | −5.7 | −8.6 |
| 6 | −22.5 | −25.2 | −18.7 | −20.8 | −9.5 | −13.2 | −18.2 | −7.5 | −2.4 | −3.9 |
| 7 | −22.4 | −23.6 | −18.6 | −19.3 | −15.3 | −12.3 | −20.3 | −6.5 | −6.2 | −7.5 |
| 8 | −24.6 | −25.4 | −17.6 | −18.2 | −12.1 | −13.9 | −19.7 | −10.2 | −4.7 | −7.6 |
| 9 | −23.5 | −25.2 | −18.7 | −21.8 | −10.5 | −13.2 | −18.2 | −7.5 | −3.4 | −3.9 |
| 10 | −22.4 | −23.6 | −18.6 | −18.3 | −14.3 | −11.3 | −19.3 | −6.5 | −5.2 | −7.5 |
| 11 | −25.6 | −25.4 | −18.6 | −18.2 | −13.1 | −13.9 | −19.7 | −10.2 | −4.7 | −7.6 |
| 12 | −23.5 | −25.2 | −18.7 | −21.8 | −9.5 | −12.2 | −17.2 | −6.5 | −3.4 | −3.9 |
| 13 | −23.4 | −23.6 | −19.6 | −18.3 | −14.3 | −11.3 | −19.3 | −7.5 | −5.2 | −7.5 |
| 14 | −24.6 | −25.4 | −18.6 | −17.2 | −12.1 | −12.9 | −18.7 | −9.2 | −3.7 | −6.6 |
| 15 | −23.5 | −25.2 | −18.7 | −22.8 | −10.5 | −12.2 | −17.2 | −7.5 | −3.4 | −3.9 |
| ... | ... | ... | ... | ... | ... | ... | ... | ... | ... | ... |
| 40 | −22.4 | −21.6 | −19.6 | −19.3 | −15.3 | −10.3 | −18.3 | −7.5 | −4.2 | −7.5 |
| 41 | −25.6 | −25.4 | −17.6 | −17.2 | −12.1 | −11.9 | −18.7 | −10.2 | −3.7 | −6.6 |
| 42 | −21.5 | −25.2 | −17.7 | −22.8 | −9.5 | −11.2 | −17.2 | −7.5 | −2.4 | −2.9 |
| 43 | −23.4 | −21.6 | −19.6 | −19.3 | −15.3 | −10.3 | −18.3 | −7.5 | −4.2 | −7.5 |
| 44 | −25.6 | −25.4 | −16.6 | −16.2 | −11.1 | −11.9 | −18.7 | −10.2 | −3.7 | −5.6 |

The third process involved the Granger causality test. As presented in Table 14 below, all five types of independent variables for each branch operation revealed significant causal relationships with RVN. At the 5% significance level, both nearby and post-integrated points, as well as closed points, demonstrated significant results for Granger causality from the <0.01 to 0.05 levels. In sum, the unit root test and Granger causality results confirm that the five independent variables for each branch management type have a statistically significant effect on the dependent variable, RVN.

**Table 14.** Granger causality test.

| Independent Var. | | RVN | Result |
|---|---|---|---|
| Closed Branch | CST | 0.04 | Sig ** |
| | SCD | 0.02 | Sig ** |
| | STF | 0.03 | Sig ** |
| | OMC | 0.05 | Sig ** |
| | RPC | 0.01 | Sig *** |
| Nearby & Post-integrated Branch | CST | 0.03 | Sig ** |
| | SCD | <0.01 | Sig *** |
| | STF | 0.02 | Sig ** |
| | OMC | 0.03 | Sig ** |
| | RPC | <0.01 | Sig *** |

Note: Granger causality $p$-values of the joint Wald statistics (** $p < 0.05$, *** $p < 0.01$).

The fourth step involved using GIRFs, which show how all variables in the model respond over time when a certain size of impact is applied to a variable in the model, based on the estimated coefficient of VAR. This tracks the effect of the unit "1-standard deviation shock" on one of the error terms for the present and future values of the endogenous variable. As depicted in Table 15, the difference in speed, reflecting the performance of bank profits of closed branches and nearby and post-integrated branches, is markedly different. In other words, both closed branches, and nearby and post-integrated ones have the same significance in reflecting the impact of each independent variable on the rate of return, but there is a significant difference in the time-series speed of reaching the rate of return. That is, the average wear-in times were 3.52 and 1.88 for closed, and nearby and post-integrated branches, respectively, which means that nearby and post-integrated branches are reflected in revenue faster than closed branches; this also implies that their shock sensitivity is much higher.

**Table 15.** The test for generalized impulse response functions (GIRFs).

| Independent Var. | | RVN | | |
|---|---|---|---|---|
| | | Immediate | Accumulative | Wear-In Time |
| Closed Branch | CST | 0.009 | 0.358 ** | 3.6 |
| | SCD | 0.013 ** | 0.421 ** | 2.9 |
| | STF | 0.008 | 0.377 ** | 3.5 |
| | OMC | 0.011 ** | 0.394 ** | 4.9 |
| | RPC | 0.014 ** | 0.449 *** | 2.7 |
| Nearby & Post-integrated Branch | CST | 0.026 *** | 0.574 *** | 1.9 |
| | SCD | 0.032 *** | 0.617 *** | 1.6 |
| | STF | 0.024 *** | 0.544 *** | 2.1 |
| | OMC | 0.023 ** | 0.517 *** | 2.3 |
| | RPC | 0.036 *** | 0.628 *** | 1.5 |

Note: The coefficients of firm value are percentage values (** $p < 0.05$, *** $p < 0.01$).

In the last stage, as seen in Table 16, we performed GFEVD for closed, nearby, and post-integrated branches using the parameters of the estimated VAR model, which measures the extent to which the time series of specific variables is explained by unexpected changes in other variables, as described in the equation below. Through this, it is possible to compare and evaluate the influence of other variables on the change in a specific variable, so that it is possible to gauge how sensitively each independent variable reacts to changes in another variable. In addition, the GFEVD, measured in this way, plays a role in providing variable factors for the operational method that each branch should take strategically. As outlined

in the equation below, the effectiveness of the two variables can be compared using the covariate ratio of the product of each variable.

$$\theta_{ij}(t) = \frac{\sum_{k=0}^{t} \left(\varphi_{ij}(k)\right)^2}{\sum_{k=0}^{t} \sum_{j=0}^{m} \left(\varphi_{ij}(t)\right)^2} \, (i, \, j = 1, 2, ..., m)$$

**Table 16.** The test for generalized forecast error variance decomposition (GFEVD).

| Year | Establishment | RVN |
|---|---|---|
| Closed Branch | CST | 2.73 |
| | SCD | 2.94 |
| | STF | 2.47 |
| | OMC | 2.31 |
| | RPC | 2.81 |
| Nearby & Post-integrated Branch | CST | 3.11 |
| | SCD | 3.46 |
| | STF | 3.28 |
| | OMC | 3.17 |
| | RPC | 3.76 |
| Sum of 5 vars for Closed Brs. << Sum of 5 vars for Nearby & Integrated Brs. | | |
| Kruskal–Wallis Statistics | | 7.485 *** |
| F-statistics | | 9.177 *** |

Note: Generalized Forecast Error Variance Decomposition (*** $p < 0.01$).

GFEVD allows for a more objective analysis compared to previously described GIRFs. In other words, it is possible to compare the objective influence on the concept representing the variable by conducting a goodness-of-fit test (7.485 ***) using Kruskal–Wallis statistics on closed branches, and nearby and post-integrated branches. As depicted in Table 16, we can check the model fit using the F-test (9.177 ***). We can check the effect of each independent variable on the dependent variable using an objectified percentage index ($\theta_{ij}(t)$). Further, we can intuitively check the sensitivity applied to the rate of return through a direct comparison of the operational methods of closed branches, and nearby and post-integrated branches.

As portrayed above, based on the results of the time-series analysis using the VAR model, we rejected the second hypothesis and confirmed that the operational methods of nearby and post-integrated branches were likely to more positively impact yield than those of the closed branches.

## 5. Conclusions

This study empirically examined the positive and negative effects of multi-channel expansion on customers based on a large amount of actual data from South Korean commercial banks. Our results were as follows. First, branch closures and deterioration in customer service due to various multiple channels had little effect on youth and young people, and active customers, but had a very negative impact on middle-aged and elderly people, and inactive customers. Second, contrary to these concerns, we confirmed that an individual's contribution to a bank based on absolute size is somewhat large for active customers who are young and young people, while the impact is low for inactive middle-aged and elderly customers. Lastly, we verified that the number of active middle-aged and elderly customers was growing at a very rapid pace before and after branch closures. In other words, the diffusion of technology occurs continuously and flexibly in all age groups. Overall, we proved that branch closures and multi-channel effects in the financial industry—the biggest practical significance of this study—are gradually being resolved by immediate technology acceptance, contrary to great concern among the public.

This study has practical and academic significance. Existing empirical studies related to multi-channel banking have many limitations regarding the use of vast practical data. It is especially difficult to utilize large-capacity individual performance data under the strict rules for personal credit information protection introduced in recent years. Moreover, from a bank's perspective, the academic use of such internal data is classified as confidential in terms of policy, unlike in the past, and is often perceived negatively. Nevertheless, we directly compared closed branches with nearby and post-integrated branches, set up several social divergence points as variables, created hypotheses, and conducted time-sequential tests. Despite the study's many practical and academic contributions, we cannot help but observe its endogenous limitations. First, we did not consider the geographic selection of the bank branches. Therefore, an approach that considers the geographical specificity of each branch is necessary. In other words, we anticipate that new independent research will be possible with map-based GIS research, along with optimization theory. Second, the selection of closed branches can be linked to social issues, such as labor quality or employee livelihood, beyond simple profit generation or geographic issues, but we have deferred research on these areas. We believe that further research is necessary in the future. Third, we did not differentiate between online banking and mobile banking. Although these styles of banking remain different, the availability of data on online banking has been declining; hence, we analyzed it by grouping it into a concept called activation. Lastly, we did not consider the transaction cost reduction effect, which is the biggest advantage of branch closure. We acknowledge that the collection of sensitive internal data is difficult in practice. We hope that this finding will be verified academically through future research.

**Author Contributions:** Conceptualization, Z.L.; Data curation, J.K.; Formal analysis, S.H. and J.K.; Funding acquisition, J.K.; Investigation, S.C. and Z.L.; Methodology, S.H. and J.K.; Project administration, J.K.; Resources, S.C.; Software, S.H. and J.K.; Supervision, Z.L.; Validation, J.K.; Writing—original draft, S.C., Z.L., S.H. and J.K.; Writing—review & editing, J.K. All authors have read and agreed to the published version of the manuscript.

**Funding:** This work was supported by the National Research Foundation of Korea (NRF) grant funded by the Korea government (No. RS-2022-00166934).

**Data Availability Statement:** Not applicable.

**Conflicts of Interest:** The authors declare no conflict of interest.

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
