# Peer review of "Determinants of Bank Closures: What Ensures Sustainable Profitability in Mobile Banking?"

_electronics, doi:10.3390/electronics12051196_

Round 1

Reviewer 1 Report

Dear Authors,

First of all, congratulation for your research. I believe that your paper brings a number of relevant information in relation to a topic of interest.

I think, however, that a number of clarifications could increase the value of the article. Thus:

- it would be useful to explain how you obtained the data used in the research, given the level of detail

- it would be relevant to statistically describe the dataset used for the research

-  population segmentation by age is not very clear. You mention twice "(1) youth and (2) young people under 55, (3) middle-aged, and (4) elderly people over age 65". Middle-aged people are those between 55-65? Does not correspond to any classification... It doesn't even seem relevant to the research approach to segment population in these brackets. It must be clarified.

- also, regarding the research design, why did you segment population into 4 age bracket, if the research hypotheses only consider 2 age ranges?

- I also believe that a substantial update and addition of the references used would increase the value of the article

I look forward to receiving the improved paper.

Best regards,

Author Response

Please refer to the attached file below.

Reviewer 2 Report

After reading this article, I noticed a number of aspects related to:

Abstract. The authors do not specify clearly what is the main objective of their research, the methodology used, the results and the conclusions obtained. I suggest the authors to realize these aspects.

Introduction. In this section, the authors do not clearly specify the main objective of the research and the most important aspect that is the gap covered in the literature by the study carried out by them.

Literature review. The authors do not describe the components shown in Figure 1 and Figure 2. I suggest the authors to realize these aspects by mentioning the related bibliographic notes.

Research methodology. The authors describe in detail the research methodology used and the research hypotheses that were the basis of the study.

Results. In this section, the authors test the research hypotheses launched and present the results in the form of tables. I did not find the section dedicated to Discussions based on the analyzed data and their interpretation. I suggest the authors to redo the Discussions section and to present the ones suggested above, comparing them with other results of specialists in the field.

Conclusions. The authors briefly present their own contributions to the realization of this study and some of the limitations of their study. I suggest the authors to develop their own contributions specifying the main economic and managerial implications, including future research directions.

Author Response

(The authors gave the same response as above.)

Reviewer 3 Report

The article addresses the problem of banking activity transformation in terms of digitalization. This problem, as convincingly argued by the authors with the involvement of statistical data on Korean banks (see Introduction), is relevant and deserves scientific research, which is confirmed by the analysis of the literature (section "Background").

The article is written intelligibly and well visualized (contains two figures and sixteen tables). The research material is well structured (see lines 71-99). The conclusions are convincingly reasoned, which is confirmed by the correct application of empirical research methods on the example of 88 bank branches, including 44 closed and 44 continued operations during the study period.

The conclusions are consistent with the purpose, objectives and results of the research. However, we recommend that the scientific novelty of the research be formulated more clearly in the abstract and in the introduction.

In our opinion, the article corresponds to the subject of the journal and deserves publication, taking into account small adjustments, including it would be appropriate to formulate the novelty of the research.

Author Response

(The authors gave the same response as above.)

Round 2

Reviewer 1 Report

Dear Authors,

Thank you for your answer to my suggestions.

I consider this paper suitable for publication.

Good luck with your future research!

Best regards,

Author Response

I appreciate your comment below. 

Your support has been a great strength to me.

Thank you, again.

Reviewer 2 Report

After reading the article, I noticed the following aspects related to:

Abstract. The authors do not clearly specify the main objective of their research, the results and the conclusions obtained. I suggest the authors to realize this.

Introduction. The authors specify the main objective of the research, which is not found in the abstract of the article. I suggest the authors to realize this and that there are no more confusions about it.

Literature review. This section still needs to be improved because some current studies cannot be found. I suggest the authors to realize this.

Author Response

We are very grateful for your nice comments.
I think that the parts that need to be improved in your proposed Abstract, Introduction, and Literature will serve as a model for our continued research in the future.
In addition, your support and comments have been a great source of strength for us.
Thank you again.